# Correlation of Pancreatic T1 Values Using Modified Look-Locker Inversion Recovery Sequence (MOLLI) with Pancreatic Exocrine and Endocrine Function

**DOI:** 10.3390/jcm9061805

**Published:** 2020-06-10

**Authors:** Norihiro Ashihara, Takayuki Watanabe, Satoko Kako, Yasuhiro Kuraishi, Makiko Ozawa, Shohei Shigefuji, Keita Kanai, Yoko Usami, Akira Yamada, Takeji Umemura, Yasunari Fujinaga

**Affiliations:** 1Department of Medicine, Division of Gastroenterology and Hepatology, Shinshu University School of Medicine, Matsumoto 390-8621, Japan; wat1400@shinshu-u.ac.jp (T.W.); ksatoko55@shinshu-u.ac.jp (S.K.); yasuhiro@shinshu-u.ac.jp (Y.K.); makitano@shinshu-u.ac.jp (M.O.); drunkenstein3@gmail.com (K.K.); tumemura@shinshu-u.ac.jp (T.U.); 2Department of Laboratory Medicine, Shinshu University Hospital, Matsumoto 390-8621, Japan; shigeto77@shinshu-u.ac.jp (S.S.); yoko89@shinshu-u.ac.jp (Y.U.); 3Department of Radiology, Shinshu University School of Medicine, Matsumoto 390-8621, Japan; coupdecoeur307@gmail.com (A.Y.); fujinaga@shinshu-u.ac.jp (Y.F.)

**Keywords:** pancreas, pancreatic exocrine insufficiency, MRI T1 mapping, modified Look-Locker inversion recovery

## Abstract

Quantifying myocardial T1 values has been useful for detecting and characterizing fibrotic appearance in myocardial infarction, focal scars, and non-ischemic cardiomyopathies. Since pancreatic exocrine function decreases with chronic pancreatic fibrosis advancement, this study examined the correlation between pancreatic T1 values and pancreatic exocrine and endocrine insufficiency. Methods: Thirty-two patients underwent abdominal contrast-enhanced MRI in our department between October 2017 and February 2019. We evaluated the T1 values of the pancreas using a modified Look-Locker inversion recovery sequence (MOLLI), pancreatic exocrine insufficiency (PEI) by fecal elastase 1 (FE1) values, and pancreatic endocrine insufficiency using fasting insulin and blood glucose levels to calculate the HOMA-β. This trial is registered in the UMIN Clinical Trials Registry as UMIN 000030067. Results: The median cohort (9 males and 23 females) age was 71 (range: 49–84) years. Eighteen patients had pancreatic cysts, three had alcohol-induced chronic pancreatitis, three had pancreatic cancer, and eight possessed other pancreatic features (two patients each with autoimmune pancreatitis, acute pancreatitis, or a bile duct tumor, one with idiopathic chronic pancreatitis, and one healthy control with negative findings). The median pancreatic T1 value measured by the MOLLI was 857.5 ms (597–2569). A significant negative correlation was found between the T1 mapping and FE1 values (r = 0.69, *p* < 0.01), with none for the T1 with HOMA-β or serum albumin, triglycerides, or body mass index. Conclusions: the pancreatic T1 values correlated significantly with pancreatic exocrine function and might be useful in PEI diagnosis.

## 1. Introduction

Pancreatic exocrine dysfunction is often encountered in patients with advanced pancreatic fibrosis, although detecting abnormalities in the absence of an advanced disease status or other symptoms is difficult. Patients without advanced pancreatic fibrosis often progress to chronic pancreatitis (CP) [1]. Accordingly, the prompt diagnosis of asymptomatic pancreatic exocrine dysfunction is important. The long-term follow-up of pancreatic cystic neoplasms [2,3] and autoimmune pancreatitis [4,5] also requires the diagnosis of pancreatic exocrine dysfunction, since these diseases cause pancreatic fibrosis and impaired exocrine function, similar to those in chronic pancreatitis.

Pancreatic exocrine dysfunction not only reduces quality of life due to fatty stools and diarrhea [6,7], but also increases the risk of complications due to infection [8], osteoporosis [9,10], cardiovascular lesions [11,12], and nutritional disorders [13]. As a result, the life expectancy of patients with pancreatic exocrine dysfunction is approximately 15–20 years less than that of healthy individuals [8,14].

Standard exocrine function tests include the secretin test as a direct measure and the BT-PABA test, benzoyl-L-tyrosyl (l-13C) alanine breath test, and fecal elastase 1 (FE1) test as indirect measures. Imaging studies have also reported that MRI/ Magnetic resonance cholangiopancreatography (MRCP) assessment of the amount of digestive fluid secreted into the intestine, after the administration of secretin, could predict pancreatic exocrine function [15,16]. However, the above tests can be time consuming.

An MRI scanning of the pancreas and biliary tract is less invasive than the ndoscopic retrograde cholangiopancreatography (ERCP) and Endoscopic ultrasound (EUS) [17,18] procedures, and so clinicians often use MRI to assess pancreatic ducts and tumors. The number of MRI examinations is increasing worldwide [18].

Recently, several reports [19,20] have described the utility of MRI T1 mapping to assess myocardial fibrosis. MRI T1 mapping is a technique for evaluating lesions by quantifying the T1 values of the target tissue. In particular, MRI T1 mapping can reflect the status of the extracellular matrix, including extracellular collagen that may be an indicator of fibrosis [21]. Similarly to myocardial fibrosis, the growth of the extracellular matrix, along with extracellular collagen in CP, causes fibrosis of the pancreatic parenchyma [22]. 

For the above reasons, we hypothesized that MRI T1 mapping could also evaluate pancreatic fibrosis and pancreatic exocrine dysfunction as an easier and more convenient method to assess pancreatic status. As pancreatic function decreases with chronic pancreatic fibrosis advancement, this study aimed to clarify the correlations between pancreatic T1 values and exocrine and endocrine insufficiency.

## 2. Methods

### 2.1. Patients

Thirty-two patients (23 females and 9 males; mean age: 71 years, range: 49–84 years) scheduled to undergo diagnostic abdominal contrast-enhanced MRI between October 2017 and January 2019 in the Department of Gastroenterology at Shinshu University School of Medicine were included in this study. The enrolled patients were all over 20 years of age and provided written consent to participate in this study (Table 1).

As the purpose of this study was to evaluate the correlation between pancreatic T1 values and pancreatic exocrine and endocrine insufficiency in various diseases, no specific illnesses were restricted. We excluded cases in which we could not obtain consent to participate in the study, cases where it was difficult to carry out MRI examinations due to such reasons as difficulty in maintaining posture, and cases where proper stool collection was difficult because of diarrhea or constipation.

Eighteen patients had pancreatic cysts, 3 had alcohol-induced CP, 3 had pancreatic cancer, and 8 possessed other pancreatic features (autoimmune pancreatitis: 2, acute pancreatitis: 2, bile duct tumor: 2, idiopathic CP: 1, negative findings (no pancreatic disease found): 1). The 2 patients with acute pancreatitis underwent MRI for the purpose of follow-up after treatment. The 2 bile duct tumors were benign and did not cause obstructive jaundice. All pancreatic cancers were located in the body of the pancreas, were less than 2 cm, and had no lymph node or distant metastases. No patients with pancreatic cancer exhibited obstructive pancreatitis. 

Fifteen patients displayed no symptoms, 10 had abdominal pain, and 7 had other symptoms (weight loss: 3, anorexia: 2, heartburn: 1, fever: 1).

### 2.2. Pancreatic Exocrine and Endocrine Function and Serological Analysis

We evaluated pancreatic exocrine insufficiency (PEI) using FE1 values [23,24]. Patient fecal samples were collected within 1 month of the abdominal contrast MRI. Pancreatic enzymes were not administered within 3 days of the stool collection in any case. Stool samples were cryogenically stored immediately after collection. FE1 was measured using a BIOSERV Pancreatic Elastase ELISA kit (Bioserv Rostock, Germany). Based on previous reports, we defined the values of <200 μg/g as mild PEI and <100 μg/g as severe PEI [25].

Pancreatic endocrine insufficiency was evaluated using fasting insulin and blood glucose levels to calculate the HOMA-β [26]. Two patients were administered subcutaneous insulin, from whom the HOMA-β could not be determined reliably.

Hemoglobin A1c (HbA1c) was quantified for the evaluation of blood glucose control. Body mass index, serum albumin, and triglycerides were assessed for the nutritional status determination.

### 2.3. MRI T1 Mapping

The T1 mapping of the pancreas using the MOLLI was performed with a 3.0 T MR scanner (MAGNETOM Vida, Siemens Healthcare GmbH, Erlangen, Germany) at the following settings: repetition time = 1500 ms, echo time = 1.47 ms, flip angle = 35 degrees, and inversion time = 127, 207, 1627, 1707, 3127, 3207, 4627, and 6127 ms. The slice thickness, matrix size, and field of view were 4 mm, 512 × 436 pixels, and 360 × 307 mm, respectively. The region of interest was placed in the pancreas on the obtained T1 map by a board-certified abdominal radiologist with 20 years of experience in diagnostic radiology. Measurements were performed in triplicate at different anatomical locations (the pancreatic head, body, and tail). The mean T1 values were used for the analysis.

### 2.4. Statistical Analysis

Statistical analyses were performed using Stat Flex version 7.0 (Artech Co., Osaka, Japan) and IBM SPSS statistical software for Windows, version 25.0 (IBM Corp., Armonk, NY). A *p*-value of <0.05 was considered statistically significant.

### 2.5. Ethics

The present study was approved by the Ethics Committee of Shinshu University School of Medicine (Matsumoto, Japan). The protocol of this investigation was conducted in accordance with the principals outlined in the Declaration of Helsinki of the World Medical Association and was approved by the Ethics Committee of Shinshu University School of Medicine (No. 3838). Written informed consent was obtained from each subject after a full explanation of the study. This trial is registered in the UMIN Clinical Trials Registry as UMIN 000030067.

## 3. Results

### 3.1. Pancreatic Exocrine and Endocrine Function and MRI T1 Mapping

The median FE1 was 350.5 µg/g (5.5–706.5). The FE1 was <200 µg/g in 10 cases (31.3%) and <100 µg/g in 3 cases (9.4%). To evaluate pancreatic endocrine function, the HOMA-β was measured in 30 cases after the exclusion of 2 cases which were subcutaneously administered insulin. The median HOMA-β was 38.57% (4.0–153). Ten patients (33.3%) had a HOMA-β of less than 30%, which indicated decreased insulin secretion in the cohort. Two cases (6.7%) had a HOMA-β of less than 15% and were considered as having a serious insulin deficiency [27] (Table 2).

The median HbA1c was 5.7% (4.3–10.0), with 6 cases (19.4%) exhibiting more than 6.5%. The HbA1c of the 2 patients who received subcutaneous insulin was 7.1% and 8.4%, respectively.

The median values of the other parameters were as follows: albumin = 4.3 g/dL (2.8–5.0), triglycerides = 101.5 mg/dL (2.5–161.0), and body mass index = 22.0 (17–30). The median pancreatic T1 value measured by MOLLI was 857.5 ms (597–2569). 

### 3.2. Correlation of Pancreatic T1 Values with Pancreatic Exocrine and Endocrine Function 

The severe PEI cases had significantly higher pancreatic T1 values than the mild PEI cases (1003.0 ms vs. 838.5 ms; *p* = 0.009). A significant negative correlation was observed between the FE1 and pancreatic T1 values (*r* = −0.715, *R*^2^ = 0.512, *p* < 0.001) (Figure 1).

In the receiver operating characteristic (ROC) curve analysis for a mild PEI diagnosis (FE1 < 200 μg/g), a pancreatic T1 value of 906.3 ms reached the maximum area under the ROC curve of 0.79318 (95% confidence interval (CI): 0.70872–0.87764; sensitivity/specificity: 0.793). In the ROC curves of severe PEI parameters (FE1 100 μg/g), a pancreatic T1 value of 1097 ms reached the maximum area under the ROC curve of 0.97701 (95% CI: 0.9494–0.9999; sensitivity/specificity: 0.930).

### 3.3. Correlation of Clinical Findings with Pancreatic Exocrine Function and Pancreatic T1 Values

The median FE1 values for each disease were as follows: pancreatic cyst, 345 μg/g; alcohol-induced CP, 370.47 g/g; pancreatic cancer, 322.27 μg/g; autoimmune pancreatitis, 438.07 μg/g; idiopathic CP, 138.14 μg/g; acute pancreatitis, 535.11 μg/g; and other pancreatic features, 477.23 μg/g. There were no remarkable differences among the results (*p* = 0.434) (Table 3).

The median T1 values for each disease were as follows: pancreatic cyst, 930.5 ms; alcohol-induced CP, 854 ms; pancreatic cancer, 817 ms; autoimmune pancreatitis, 1306.5 ms; idiopathic CP, 984.0 ms; acute pancreatitis, 725 ms; and other features, 829 ms. No significant differences were noted among the results (*p* = 0.085).

In comparisons of the symptoms, the median FE1 displayed the following: heartburn, 220 μg/g; abdominal pain, 498.12 μg/g; asymptomatic, 231.00 μg/g; anorexia, 575.10 μg/g; weight loss, 331.07 μg/g; and fever, 5.50 μg/g (*p* = 0.072). The median T1 displayed the following: heartburn, 940 ms; abdominal pain, 861.5 ms; asymptomatic, 859.0 ms; anorexia, 722.5 ms; weight loss, 829 ms, and fever, 2569 ms (*p* = 0.377).

No remarkable association was seen for the HOMA-β and pancreatic T1 values (*r* = −0.094; *R^2^* = 0.001; *p* = 0.636) (Figure 2), nor were any correlations noted for albumin, triglycerides, or body mass index.

## 4. Discussion

In this study, a significant correlation was identified between the pancreatic MRI T1 mapping values and pancreatic exocrine function. Several earlier reports have examined the correlation between pancreatic MRI/MRCP findings and pancreatic exocrine status. Madzak et al. observed that the measurement of intestinal fluid volume using MRI after secretin administration was useful for evaluating pancreatic exocrine function in cystic fibrosis patients and healthy controls [15]. In their assessment of pancreatic MRI T1 mapping [28], Tirkes et al. measured pancreatic values in 45 patients with mild CP and 53 healthy controls to identify higher T1 mapping values in the mild CP group. This indicated the utility of pancreatic MRI T1 mapping in CP diagnosis, which was in agreement with our findings.

Pancreatic exocrine dysfunction is thought to be caused by a progressive fibrosis of pancreatic parenchyma and the loss of acinar cells. Myocardial T1 values have already been measured clinically and shown as useful for the quantitative and visual assessment of the extent of myocardium damage by myocardial lesions [19,20]. A histopathological study using myocardial biopsy samples uncovered a correlation between myocardial fibrosis and T1 values [29]. Hence, we hypothesized that the measurement of pancreatic T1 values could estimate pancreatic parenchyma fibrosis and pancreatic exocrine function. This notion was confirmed by the correlation between the two parameters. 

In contrast, no significant associations were witnessed between the pancreatic T1 values and pancreatic endocrine function in the cohort. Pancreatic endocrine dysfunction in CP usually occurs with highly advanced pancreatic parenchymal fibrosis [30] and therefore following pancreatic exocrine dysfunction. In other words, our results showed that the pancreatic T1 values correlated with the exocrine dysfunction caused by pancreatic parenchymal fibrosis, and not with endocrine secretion caused by decreased islet cells. However, there were only 2 cases of a HOMA-β of less than 15%, which indicated severe insulin deficiency in Japanese patients; the possibility of a correlation between pancreatic endocrine dysfunction and T1 mapping values may not be completely ruled out. If more cases of advanced pancreatic endocrine dysfunction associated with the progression of fibrosis are included in future studies, a relationship between T1 values and pancreatic endocrine function may appear.

## 5. Limitations

This study was a prospective observational study with limited patient choice that contained a small number of participants. Unfortunately, we had a lack of control over this. Moreover, the influence of pancreatic fat content was not fully considered, and no other criteria were used to evaluate exocrine pancreatic function.

## 6. Conclusions

A significant negative correlation was observed between the pancreatic T1 values and pancreatic exocrine function, suggesting the possibility of estimating pancreatic exocrine status by pancreatic T1 mapping.

## Figures and Tables

**Figure 1 jcm-09-01805-f001:**
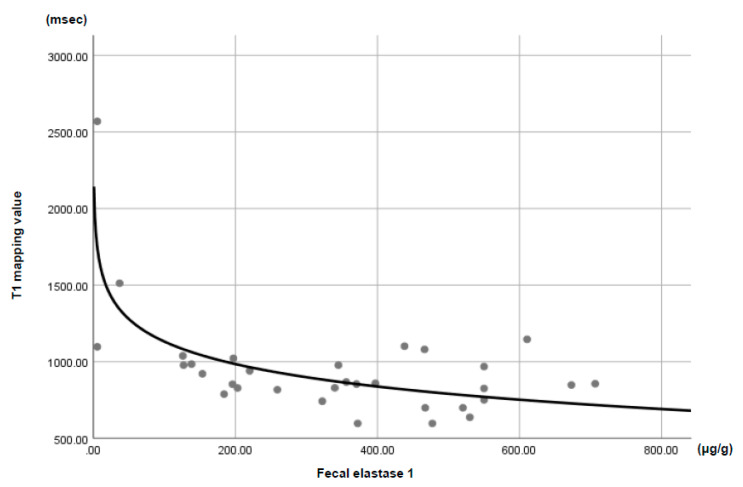
A significant correlation was seen between the T1 mapping values and fecal elastase 1 (*r* = −0.715; *R^2^* = 0.512; *p* < 0.001).

**Figure 2 jcm-09-01805-f002:**
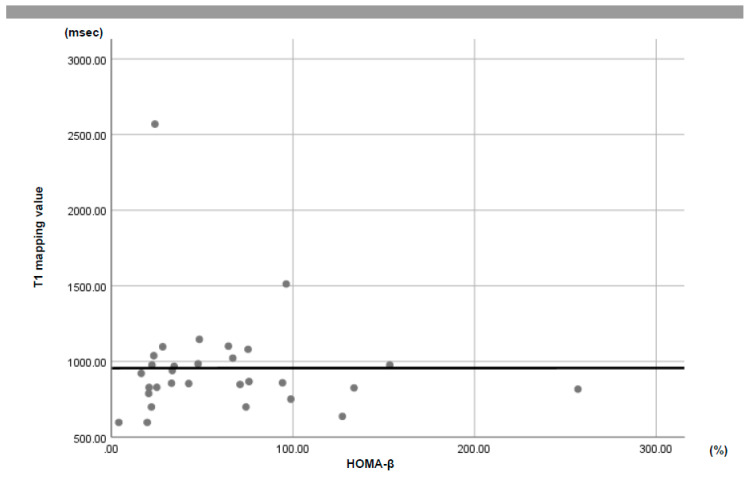
No remarkable correlation was observed between the T1 mapping values and HOMA-β (*r* = −0.094, *R^2^* = 0.001, *p* = 0.636).

**Table 1 jcm-09-01805-t001:** Patient characteristics.

		N = 32
Age, years (median, range)		71, 49–84
Gender (male, female)		9, 23
Disease		
Pancreatic cysts		
	IPMN	12
	Serous cystic neoplasm	1
	Simple cyst	5
Alcohol-induced CP		3
Idiopathic CP		1
Pancreatic cancer		3
Autoimmune pancreatitis		2
Acute pancreatitis		2
Bile duct tumor		2
Negative findings		1

**Table 2 jcm-09-01805-t002:** Laboratory characteristics.

	N = 32 (Average, Range)
Total protein (g/dL)	7.1 (5.7–8.4)
Albumin (g/dL)	4.2 (2.8–5.0)
Triglycerides (mg/dL)	99.8 (39–161)
Amylase (U/L)	107.0 (45.0–296.0)
P-amylase (U/L)	57.0 (5.0–239.0)
HbA1c (%)	6.0 (4.3–10.0)
Fasting blood sugar (mg/dL)	114.7 (88–290)
Fasting insulin (μU/mL)	11.85 (1.5–162.0)
Blood elastase-1 (ng/dL)	250.2 (2.5–1300.0)
Fecal elastase-1 (μg/g)	361.7 (5.5–706.5)

**Table 3 jcm-09-01805-t003:** Pancreatic exocrine function and pancreatic T1 values in each disease.

	Fecal Elastase-1 (ug/g)		T1 Values (ms)	
Pancreatic cysts	345	NS *p* = 0.434	930.5	NS *p* = 0.085
Alcohol-induced CP	370.47		854	
Idiopathic CP	138.14		984	
Pancreatic cancer	322.27		817	
Autoimmune pancreatitis	438.07		1306.5	
Acute pancreatitis	535.11		725	
Others	477.23		829	

NS; Not Significant

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
