# Peer review of "Correlation of Pancreatic T1 Values Using Modified Look-Locker Inversion Recovery Sequence (MOLLI) with Pancreatic Exocrine and Endocrine Function"

_jcm, 2020, doi:10.3390/jcm9061805_

Round 1

Reviewer 1 Report

The manuscript describes and interesting concept of using T1 MRI mapping to predict pancreatic exocrine and endocrine functions. However, it would be to expand the background information on what exactly can be detected by T1 mapping, in which situation it is used, etc.

It appears that the study was not well designed and it is missing controls - there was only one control sample grouped with the others, so it is not clear if there is any difference at all between diseased pancreas versus healthy one.

Overall it was very hard to understand the significance of the work presented.

However, there are some major concerns: 

  • Since it seems that a clear correlation in pancreas between fibrosis and T1 values is not established, better experimental design is necessary with more healthy controls to prove the concept!
  • Perhaps label data from the control pancreas with a different symbol for readers to better understand where values fall for the healthy pancreas. It is crucial to assess the readouts of the healthy controls.
  • It is unclear under which conditions would patients be subjected to T1 MRI? what is the advantage over using other exocrine or endocrine function test?
  • From the introduction it is unclear what the T1 MRI mapping detects? Is it known that it correlates with fibrosis? Could for example ultrasound be used instead?

Minor issues:

It appears that some prepositions are missing in some sentences.

Line 109: if only 30 patients were assessed for HOMA-β , shouldn't percentages reflect that? (it is not clear 10 patients = 35.8% => was 30/32 patients assessed and then 2 were excluded = 30-2=28)

Author Response

Dear Reviewer 1,

Thank you for your suggestions and comments on our paper. We have attempted to address each concern to the best of our ability. All relevant changes have been highlighted in the revised version.

Point 1: It appears that the study was not well designed and it is missing controls - there was only one control sample grouped with the others, so it is not clear if there is any difference at all between diseased pancreas versus healthy one.

Response 1: Thank you for raising this point. However, we would like to clarify that our study aimed to investigate the correlation between pancreatic exocrine and endocrine function and T1 values, and not whether pancreatic T1 values were useful for the diagnosis of chronic pancreatitis.  To clarify this purpose, we have made changes in the Introduction (Line 36-42). Furthermore, the case selection criteria have been described in the Methods (Line 70-72). To avoid misunderstanding, we have reworded the term “negative control” to “negative findings”. We have specified a lack of controls in the Limitations (Line 240).

Point 2: Since it seems that a clear correlation in pancreas between fibrosis and T1 values is not established, better experimental design is necessary with more healthy controls to prove the concept!

Response 2: We appreciate this comment. Previous reports have described that FE1 correlates with pancreatic fibrosis. In our study, by demonstrating a correlation between pancreatic T1 values and FE1, we have hypothesized that pancreatic T1 values correlate with pancreatic fibrosis.

Point 3: Perhaps label data from the control pancreas with a different symbol for readers to better understand where values fall for the healthy pancreas. It is crucial to assess the readouts of the healthy controls.

Response 3: As the Reviewer points out, we agree that clear label data should be shown for pancreatic fibrosis. We have also shown the results (Line 159-163) of T1 values examined ​​using ROC curves based on FE1 values, which is a criterion for mild and severe PEI. 

Point 4: It is unclear under which conditions would patients be subjected to T1 MRI? what is the advantage over using other exocrine or endocrine function test?

Response 4: Thank you for this question. We have added a description on the usefulness of MRI examination in the Introduction (Line 52-63).

Point 5: From the introduction it is unclear what the T1 MRI mapping detects? Is it known that it correlates with fibrosis?

Response 5: Thank you for this comment. The correlation between myocardial fibrosis and T1 mapping has been proven (Line 55-60). Since pancreatic fibrosis is considered the same condition as myocardial fibrosis, we hypothesized that similar evaluations could be performed by T1 mapping (Line 61-63).

Point 6: Could for example ultrasound be used instead?

Response 6: We appreciate this question. EUS is used for ultrasound of the pancreas, but involves invasion for the test itself (Line 52-54). In addition, the results are often inconsistent depending on the skill and observation conditions of the operator. Further study is needed at present for EUS as an accurate evaluation method for exocrine pancreatic dysfunction. 

Point 7: It appears that some prepositions are missing in some sentences.

Line 109: if only 30 patients were assessed for HOMA-β , shouldn't percentages reflect that? (it is not clear 10 patients = 35.8% => was 30/32 patients assessed and then 2 were excluded = 30-2=28)

Response 7: Thank you for these comments. We have reread the manuscript carefully and changed the calculation result on Line 139-140.

Reviewer 2 Report

The topic presented about the possible correlation of pancreatic T1 values using modified Look-Locker inversion recovery sequence (MOLLI) with predicting pancreatic exocrine end endocrine functions is relevant. In fact, as reported by Authors, available diagnostic tests in particular for exocrine insufficiency diagnosis are suboptimal. Despite the interesting topic, the paper requires revisions.

  1. Paper’s hypothesis and methodology are not clear enough. In the Introduction section, the Authors report pancreatic exocrine and endocrine insufficiency as a result of the progressive fibrosis and loss of pancreatic parenchyma that occur in chronic pancreatitis. They want to examine the correlation between pancreatic T1 values and decrease of the pancreatic function due to chronic pancreatic fibrosis. However, as reported in the Methods section, patients included in the study had various pancreatic diseases (lines 67-69: pancreatic cysts, pancreatic cancer, acute pancreatitis, bile duct tumor, autoimmune pancreatitis). As a matter of fact, only five patients are reported to be affected by chronic pancreatitis. I do not understand the inclusion criteria of the study and the reason why different diseases (that behave differently and have a different potential effects on pancreatic parenchyma) were included. It is true that only few other papers dealing with this methodology have been published in the Literature (see Tirkes T, Lin C, Fogel EL, Sherman SS, Wang Q, Sandrasegaran K. T1 mapping for diagnosis of mild chronic pancreatitis. J Magn Reson Imaging. 2017 Apr;45(4):1171-1176) but in previous report (such as in Tirkers et al paper) patients were excluded from the study if they had acute pancreatitis, cystic or solid pancreatic neoplasm (only chronic pancreatitis were included). I think that the study results could be affected by the patients selection.
  2. Furthermore, clinical criteria for patients selection are not reported. Were all the patients asymptomatic? A part from fecal elastase 1 (that has some well-known limitations), has the exocrine pancreatic function been established or evaluated with other criteria?
  3. The Authors report a significant negative correlation between pancreatic T1 values and pancreatic exocrine function, suggesting the possibility of estimating pancreatic exocrine status by pancreatic T1 mapping. However, no clinical correlation is reported in the paper. I think it would be interesting to give some suggestions on the possible timing of T1 mapping in patients with chronic pancreatitis: early in the disease course? At the diagnosis?
  4. Finally, there are some typos that should be correct (see for example line 116-117)

Author Response

Dear Reviewer 2,

Thank you for your suggestions and comments on our report. We have attempted to address each concern to the best of our ability. All relevant changes have been highlighted in the revised version.

Point 1: Paper’s hypothesis and methodology are not clear enough. In the Introduction section, the Authors report pancreatic exocrine and endocrine insufficiency as a result of the progressive fibrosis and loss of pancreatic parenchyma that occur in chronic pancreatitis. They want to examine the correlation between pancreatic T1 values and decrease of the pancreatic function due to chronic pancreatic fibrosis. However, as reported in the Methods section, patients included in the study had various pancreatic diseases (lines 67-69: pancreatic cysts, pancreatic cancer, acute pancreatitis, bile duct tumor, autoimmune pancreatitis). As a matter of fact, only five patients are reported to be affected by chronic pancreatitis. I do not understand the inclusion criteria of the study and the reason why different diseases (that behave differently and have a different potential effects on pancreatic parenchyma) were included. It is true that only few other papers dealing with this methodology have been published in the Literature (see Tirkes T, Lin C, Fogel EL, Sherman SS, Wang Q, Sandrasegaran K. T1 mapping for diagnosis of mild chronic pancreatitis. J Magn Reson Imaging. 2017 Apr;45(4):1171-1176) but in previous report (such as in Tirkers et al paper) patients were excluded from the study if they had acute pancreatitis, cystic or solid pancreatic neoplasm (only chronic pancreatitis were included). I think that the study results could be affected by the patients selection.

Response 1: We appreciate these comments on our paper. Firstly, we would like to clarify that our study aimed to investigate the correlation between pancreatic exocrine and endocrine function and T1 values, and not whether pancreatic T1 values were useful for the diagnosis of chronic pancreatitis.  To clarify this purpose, we have made changes in the Introduction (Line 36-42). Furthermore, the case selection criteria have been described in the Methods (Line 70-72).

To avoid misunderstanding, we have reworded the term “negative control” to “negative findings”. We have specified a lack of controls in the Limitations (Line 240).

 In this study, pancreatic cystic neoplasms and autoimmune pancreatitis cause fibrosis of the pancreatic parenchyma and exocrine dysfunction similarly to chronic pancreatitis, so they were not excluded.

 Contrast-enhanced MRI was performed for the purpose of scrutinizing the underlying disease, and patients that consented to this examination were targeted (Line70-72). These cases did not include active pancreatitis or obstructive jaundice (Line 82-84).

Point 2: Furthermore, clinical criteria for patients selection are not reported. Were all the patients asymptomatic?

Response 2: Thank you for this question. We have added the patients’ symptoms to the Methods (Line 85-86). The results for FE1 and T1 values have been added to the Results (Line 166-179).

Point 3: A part from fecal elastase 1 (that has some well-known limitations), has the exocrine pancreatic function been established or evaluated with other criteria?

Response 3: We appreciate this comment. In this study, all of the samples were in good condition and measurable. Unfortunately, exocrine function tests using other measurement methods were not conducted in this study. We have added this as a limitation to this study.

Point 4: The Authors report a significant negative correlation between pancreatic T1 values and pancreatic exocrine function, suggesting the possibility of estimating pancreatic exocrine status by pancreatic T1 mapping. However, no clinical correlation is reported in the paper.

Response 4: Thank you for raising this point. The correlation with clinical findings has been added to the Results (Line 166-179).

Point 5: I think it would be interesting to give some suggestions on the possible timing of T1 mapping in patients with chronic pancreatitis: early in the disease course? At the diagnosis?

Response 5: We appreciate this suggestion. However, along with our response to Point 1, this study did not examine the relationship between the duration of chronic pancreatitis and T1 mapping. 

Point 6: Finally, there are some typos that should be correct (see for example line 116-117)

Response 6: Thank you for pointing these out. We have corrected them and carefully rechecked the paper.

Reviewer 3 Report

The paper from Norihiro Ashihara et al describes the correlation of pancreatic T1 values with pancreatic exocrine and endocrine function, extrapolating its useful use quantifying myocardial fibrosis after infarction. This quantification may be useful in clinical practice, which turns this study a contribution to evaluate the prognosis of the pancreatic endocrine and exocrine function under different pathologies.

The paper is well-written and with clear objectives but it might be improved with the following suggestions:

Line 25 and 69 – Why only one normal control? Much stronger conclusions may be drawn if a control group had been studied.

There is not enough clinical data, which should be completed: Steatorrhea, jaundice, pain, etc.

Line 66 - The 18 pancreatic cysts must be specified: diagnosis, location, surgical indication or not

Line 67 – The 3 pancreatic cancer cases must be specified: staging, location, etc.

Line 70 – The sentence is not complete, and it should be added a table with all the biochemical data per patient

Author Response

Dear Reviewer 3,

Thank you for your suggestions and comments on our manuscript. We have attempted to address each concern to the best of our ability. All relevant changes have been highlighted in the revised version.

Point 1: Line 25 and 69 – Why only one normal control? Much stronger conclusions may be drawn if a control group had been studied.

Response 1: We appreciate this important comment. Firstly, we would like to clarify that our study aimed to investigate the correlation between pancreatic exocrine and endocrine function and T1 values, and not whether pancreatic T1 values were useful for the diagnosis of chronic pancreatitis.  To clarify this purpose, we have made changes in the Introduction (Line 36-42). Furthermore, the case selection criteria have been described in the Methods (Line 70-72). To avoid misunderstanding, we have reworded the term “negative control” to “negative findings”. We have specified a lack of controls in the Limitations (Line 240).

 In this study, pancreatic cystic neoplasms and autoimmune pancreatitis cause fibrosis of the pancreatic parenchyma and exocrine dysfunction as well as chronic pancreatitis, so they were not excluded.

Point 2: There is not enough clinical data, which should be completed: Steatorrhea, jaundice, pain, etc.

Response 2: Thank you for this advice. Additional patient symptoms were added to the Methods (Line 85-86). We have also added the results for FE1 and T1 values ​​in the Results (Line 166-179).

Point 3: Line 66 - The 18 pancreatic cysts must be specified: diagnosis, location, surgical indication or not

Response 3: According to the Reviewer’s comment, we have added more detailed information in Table 1. None of the cysts were operated on.

Point 4: Line 67 – The 3 pancreatic cancer cases must be specified: staging, location, etc.

Response 4: Thank you for this comment. We have added more detailed information on Line 80-84.

Point 5: Line 70 – The sentence is not complete, and it should be added a table with all the biochemical data per patient

Response 5: We appreciate these comments. The sentence has been completed and we have added Tables 1 and 2 to better characterize the cohort.

Round 2

Reviewer 1 Report

Thank you for clarifying the points raised by reviewers, and incorporating the changes.

Perhaps it would be easier to have a table with the characteristics described in the first 2 paragraphs in section: "Correlation of clinical findings with pancreatic exocrine and endocrine function and pancreatic T1 values".

Could higher resolution figures be uploaded?

At the first mention "negative findings" perhaps add a clarification in parenthesis, such as "no pancreatic disease found" or similar.

Table 1 could be improved (e.g. section "Pancreatic cysts" with categories - perhaps keep the count for each of the 3 categories, but remove 18 on the top).

Author Response

Dear Reviewer 1,

Thank you for your suggestions. We have attempted to address each concern to the best of our ability. All relevant changes have been highlighted in the revised version.

Point 1: It would be easier to have a table with the characteristics described in the first 2 paragraphs in section: "Correlation of clinical findings with pancreatic exocrine and endocrine function and pancreatic T1 values".

Response 1: I revised as requested in (Table 3).

Point 2: Could higher resolution figures be uploaded?

Response 2: I done.

Point 3: At the first mention "negative findings" perhaps add a clarification in parenthesis, such as "no pancreatic disease found" or similar.

Response 3: I done.

Point 4: Table 1 could be improved (e.g. section "Pancreatic cysts" with categories - perhaps keep the count for each of the 3 categories, but remove 18 on the top).

Response 4: I revised as requested in (Table 1).

Reviewer 2 Report

Dear Authors

I have read with interest your response and the changes you made to your paper.

I think that the introduction and the aim of your work is now more clear.

I do not have further remarks

kind regards

Author Response

Dear Reviewer 2

Thank you very much for providing important comments. We are thankful for the time and energy you expend.

Best regards, 

Reviewer 3 Report

The authors replied to all comments and improved the manuscript.

Author Response

Dear Reviewer 3  

Thank you very much for providing important comments. We are thankful for the time and energy you expend.

Best regards,